# Prevalence and Associated Factors of Dental Caries in Syrian Immigrant Children Aged 6–12 Years

**DOI:** 10.3390/children10061000

**Published:** 2023-06-02

**Authors:** Zeynep Meva Altaş, Mehmet Akif Sezerol

**Affiliations:** 1Ümraniye District Health Directorate, Istanbul 34764, Turkey; 2Epidemiology Program, Institute of Health Sciences, Istanbul Medipol University, Istanbul 34820, Turkey; masezerol@gmail.com; 3Department of Public Health, School of Medicine, Istanbul Medipol University, Istanbul 34820, Turkey; 4Health Management Program, Graduate Education Institute, Maltepe University, Istanbul 34820, Turkey; 5Sultanbeyli District Health Directorate, Istanbul 34935, Turkey

**Keywords:** immigrant children, oral health, dental caries

## Abstract

Immigrant children are among the groups that are sensitive to problems related to dental health. The aim of this study was to examine the dental caries of Syrian immigrant children. The study is a descriptive and retrospective study. Its population consists of Syrian immigrant children aged 6–12 years who were screened for dental health in the year 2022 in Istanbul. DMF-T (permanent teeth) and dmf-t (milk teeth) indices were used, which are the (t-T) criteria obtained by dividing the sum of caries (d-D), caries extracted (m-M) and caries-filled (f-F) teeth by the number of people examined. Higher dmft and DMFT scores indicate worse dental health. Dental screening was performed on 549 Syrian immigrant children. In total, 27.2% (n = 149) were brushing their teeth once a day and 97.3% of the children (n = 534) had at least one decayed tooth. The dmft score for the 6–7 year age (6.45 ± 3.33) group was significantly higher than the 8–9 year (4.98 ± 2.78) and 10–12 year (3.22 ± 2.02) age groups (*p* < 0.001). In our study, the dental caries were seen at a very-high frequency among immigrant children and the habit of tooth brushing remains at a low level. Lower age was the relevant factor for dental caries in our study.

## 1. Introduction

Oral health, which is responsible for the well-being of general health and quality of life, is also one of the priority health areas of the World Health Organization [1]. Children are among the groups that are sensitive to problems related to oral and dental health [2]. Oral and dental health problems seen in school-age children generally include dental caries, gum diseases, dislocations, trauma and injuries [3]. Dental caries, which constitute an important part of oral and dental health diseases, are a common public health problem that can negatively affect other systems [4]. The risk of dental caries is higher in children whose family members have dental caries, in those who consume a lot of sugary foods and beverages, in those who need special health care and in children who use oral apparatuses such as orthodontic treatment [5]. In addition, dental caries are related to the age and gender of the children [6]. The family income and educational level of parents are the other factors related to dental caries among children [7].

The first deciduous teeth usually erupt at 5–8 months of age, and the eruption of the deciduous teeth should be completed by 27 months [8]. In children, deciduous teeth fall out at 6–13 years of age. Permanent teeth begin to erupt after 5–6 years of age [9]. After the age of seven, the incidence of caries in permanent teeth starts to increase, especially during the school period. Studies from different countries give the results of high prevalence of dental caries in school-aged children. In a study conducted in Turkey, the prevalence of the dental caries in children aged 7–12 years was reported as 68.89% [10]. Another study conducted among 6–12-year-old children reported the prevalence of dental caries as 81.87% [11]. A study conducted in China reported that the prevalence of dental caries was 41.15% in primary and secondary school students [12]. In another study conducted in Libya, 78.0% of the first grade students had tooth decay in their primary teeth [13]. According to the data from the Centers for Disease Control and Prevention (CDC), nearly half of children between 6 and 8 years of age have had at least one decayed tooth from their deciduous teeth [14]. According to these results, the high prevalence of dental caries among school-aged children is a global public health problem that deserves attention.

The American Academy of Pediatric Dentistry recommends that the first clinical oral examination should be carried at the eruption of the first tooth and no later than 12 months. The oral examination should be repeated every six months if there is no extra special healthcare need for the child [15]. However, studies show that dentist visits for the oral health of children are not sufficient [16]. In one of the studies, more than half of the parents thought that they should be examined by the dentist in case of a dental problem in their child. In the same study, nearly two-thirds of parents had never taken their child to the dentist before and none of their children had ever received preventive dental care [17].

One of the most important habits in preventing tooth decay is tooth brushing. The habit of tooth brushing, recommended to be gained in early childhood and accepted to be among the basic health behaviors, is insufficient in our country [18]. It is inevitable that this problem is seen in immigrants, who are a disadvantaged group in accessing health services. According to studies, oral health and hygiene is negatively affected by children’s social class [19,20,21]. A study conducted in Belgium reported oral health disparities among primary school children. The rate of those with dental caries was found to be higher in children in the low-income group. In the same study, approximately 1 in 8 children did not visit the dentist in the previous 5 years [22].

Migration is defined as population movements in which people are displaced individually or collectively, regardless of their cause, structure and duration [23]. However, the causes for migration are mostly economical, social and political reasons [24]. Due to the war that started in Syria in 2011, many individuals had to migrate to Turkey. According to the latest data from the Directorate of Migration Management dated 25 May 2023, the number of registered Syrian individuals under temporary protection in Turkey is 3,411,029 and nearly half of them (1,635,397) are under the age of 18 years [25]. Thus, Turkey deserves to be a large focus of health surveys regarding Syrian immigrant children. However, there is limited number of studies examining the health status of Syrian immigrant children living in Turkey.

Migration has been accepted as an important social determinant of health [26]. Immigrants may experience some difficulties and disparities in use of health services in their places [27]. These difficulties are mostly caused by language problems, low-economic conditions, low-education levels, poor nutritional status and poor hygienic conditions [27,28]. Thus, oral health inequalities can be seen in children in different racial and ethnic groups. In children aged 2–5 years, about 33% of Mexican American and 28% of non-Hispanic Black children have had dental caries in their primary teeth based on the CDC’s data. This percentage is reported as 18% among non-Hispanic White children. Nearly 70% of Mexican American children aged 12–19 years have had dental caries in their permanent teeth, compared with the percentage of 54% among non-Hispanic White children [14].

In order to increase the access of Syrian immigrants to preventive and basic health services in our country, there are immigrant health centers (IHCs) affiliated to primary health care institutions in places where these people live intensively. IHCs offer similar services to family health centers. In addition, strengthened immigrant health centers (SIHCs) are available in places where the Syrian population is higher. SIHCs also provide health services in clinics of internal medicine, pediatrics, gynecology and obstetrics, dentistry, etc. Health services provided in SIHCs are free of charge for immigrants.

For the prevention of oral and dental diseases, the best time for intervention is childhood. For this reason, the aim of this study is to determine the dental caries of Syrian immigrant children between the ages of 6 and 12 years in the Sultanbeyli district and to examine the factors associated with dental caries.

## 2. Materials and Methods

The study was conducted with the data for Syrian immigrant children aged 6–12 years who were screened for oral and dental health in 2022 in the SIHC of the District Health Directorate of Sultanbeyli in Istanbul. Sultanbeyli is a district in Istanbul, Turkey. The Sultanbeyli district has the lowest socio-economical development index when compared to other districts of Istanbul [29]. Additionally, a high number of Syrian immigrants live in Sultanbeyli.

### 2.1. Design of the Study and Participants

The study is a descriptive and retrospective study. Its population consists of Syrian immigrant children aged 6–12 years who were screened for oral and dental health in the year 2022 from a SIHC of a District Health Directorate in Istanbul, Turkey. The screenings were carried out in the oral and dental health screening unit within SIHC, after informing the children and their families who applied to the immigrant health center for any reason. Sterilization rules were followed during the dental examination.

In the study, there were 612 applications to the oral and dental health screening unit in a one-year period. However, the number of children between 6–12 years of age was 549. All these 549 immigrant children were included in the study. There was no exclusion criteria. Written informed consent was obtained from parents for the dental examination and screening procedures of their children. Demographic data and health records of these 549 immigrant children were examined through file records retrospectively.

### 2.2. Measures

The data were obtained as a result of the dental screening performed by the same dentist with clinical experience. The dentist was independent of the study authors. The age and gender of children, treatment information, gingival bleeding history, presence of toothbrush, frequency of tooth brushing, use of interface brush and dental floss, reasons for going to the dentist before, whether there was a placeholder in the mouth, and the result of the dental examination were evaluated in the study. The dependent variable of the study was the presence of dental caries, and the aim was to examine the relationship between other variables and dental caries.

In epidemiological studies, various indices have been developed to identify dental caries and to reveal their causes and risks. The main purpose of this index system is to collect data in a similar way in order to compare data about dental caries between different societies [30]. For the evaluation of oral health, dmf-t (milk teeth) and DMF-T (permanent teeth) indices were used in our study. This indices represent the total number of decayed teeth, removed or filled teeth because of the decay [31]. The (t-T) criteria obtained by dividing the sum of caries (d-D), caries extracted (m-M) and caries-filled (f-F) teeth by the number of people examined. Thus, higher dmft and DMFT scores indicate worse dental health. DMFT score ranges between 0 and 28/32. Since dmft is the score for primary teeth, it varies between 0 and 20 points [32]. For gingival bleeding, no index was used. Parents were asked by the dentist in the course of dental screening if their child had gingival bleeding during tooth brushing.

### 2.3. Statistical Analysis

SPSS (Statistical Package for Social Sciences) for Windows 25.0 program was used for data recording and statistical analysis. The descriptive data were presented with mean, standard deviation, median, minimum and maximum values, numbers (n) and percentages (%). Conformity of continuous variables to normal distribution was examined by visual (histogram and probability charts) and analytical methods (Kolmogorov–Smirnov/Shapiro–Wilk tests). The Student *t* test was used to compare the two groups that were normally distributed, and the ANOVA test was used to compare more than two groups that have normal distribution. The comparison of the categorical data was analyzed with the Pearson chi square test. Logistic regression analysis was used as the multivariate analysis. A value of *p* ≤ 0.05 was accepted as statistically significant.

### 2.4. Ethics

Ethics committee approval was obtained from Istanbul Medipol University Non-Interventional Clinical Research Ethics Committee with the decision number 874 on 13/10/2022 for the study.

## 3. Results

In the study, dental screening was performed with 549 Syrian immigrant children aged 6–12 years. Of these, 50.3% (n = 276) were females and 49.7% (n = 273) were males. The median age was 7 years (6.0–12.0). The majority of the parents were primary school graduates: 37.7% (n = 207) (Table 1).

None of the children had a history of orthodontic treatment or use of retainers. Gingival bleeding was seen in 5.1% (n = 28) of the children. The percentage of children who had their own toothbrush was 70.9% (n = 388). Of the children, 28.3% (n = 155) were not brushing their teeth. The percentage of children brushing their teeth 2–4 times a day was 19.5% (n = 107). While 25.0% (n = 137) were brushing irregularly, 27.2% (n = 149) were brushing their teeth once a day. Only one child (0.2%) used dental floss. Of the children, 61.3% (n = 332) had not previously attended a dental examination. Prior to this, 34.9% (n = 189) of the patients went to a dental examination mostly in the presence of pain and complaints; the percentage of those who went for treatment and follow-up was 3.7% (n = 20). Only one child had an annual dental examination without complaints. The rate of children who were treated with fluoride varnish was 5.4% (n = 29) (Table 2).

When the factors that may affect the tooth brushing habits of children were evaluated, children whose parents were university graduates and who were in the 6–7 years age group were brushing their teeth at a significantly higher percentage (*p* = 0.027 and *p* = 0.005, respectively). There was no significant relationship between gender and tooth-brushing habits (*p* = 0.486). The percentage of tooth brushing was higher in children who had attended to the dentist before, but statistical significance was not observed (*p* = 0.180) (Table 3).

When the prevalence of dental caries in children was evaluated according to DMFT and dmft indexes, 97.3% of the children (n = 534) had at least one decayed tooth. Fifteen children (2.7%) had no tooth decay. Six (40.0%) of these children were male and nine (60%) were female. All except three had their own toothbrush and were brushing their teeth. Dental health behaviors and age and gender characteristics of children without caries are given in Table 4.

Factors that may be associated with DMFT and dmft scores were evaluated. The dmft score for the 6–7 year age group was significantly higher than the 8–9 year and 10–12 year age groups (*p* < 0.001). However, there was no statistically significant relationship between DMFT score and age groups (*p* = 0.158). There were no significant effects on dmft and DMFT scores of gender, child’s own toothbrush, brushing teeth at least once a day and previously visiting the dentist (*p* > 0.05). The dmft score for those whose parents were university graduates was lower than the others. Although statistical significance was not observed, the *p* value was close to the significance level (*p* = 0.053). There was no statistically significant relationship between the education status of the parents and the DMFT score (*p* = 0.725). Both gingival bleeding and fluoride varnish application have no significant effect on dmft and DMFT scores (Table 5).

For the evaluation of the factors related with dmft score, logistic regression analysis was used as the multivariable analysis. Since there was a statistically significant relationship between dmft score and age in univariate analysis, the dmft score was dichotomized as being below or above the mean value. While the dependent variable of the logistic regression model was the dmft value above the mean value, gender, age and fluoride varnish application were considered as independent variables. The reason for including these three variables as independent variables in the model is that results were found to be significant or close to the significance level in the univariate analysis. According to the results of the analysis, no significant relationship was found between gender and fluoride varnish application and dmft score. The children in the 6–7 age group had a higher dmft score by 9.863 times (95% C.I.: 5.285–18.407) compared to the 10–12 age group, while the 8–9 age group had a 4.358-fold risk (95% C.I.: 2.175–8.732) of a higher dmft score compared to the 10–12 age group (*p* < 0.001 for both) (Table 6).

## 4. Discussion

The prevalence of dental caries in childhood is increasing, especially in low- and middle-income countries, and constitutes a significant disease burden. Since almost all of the risk factors for dental caries are modifiable risk factors, the development of dental caries in childhood can be prevented with appropriate public health interventions. As with many non-communicable diseases, socio-economic, social, behavioral and environmental factors play a major role in the prevention of dental caries [33]. Since immigrants are disadvantaged groups in terms of these factors, we aimed to evaluate dental caries in immigrant children.

To prevent dental caries, oral care behaviors such as tooth brushing are extremely important. In our study, the percentage of children who had their own toothbrush was 70.9% and of the children, only 19.5% were brushing their teeth 2–4 times in a day. In a study conducted among Mexican schoolchildren aged between 6 and 12 years, the prevalence of tooth brushing (at least two times in a day) was reported as 52.8% [34]. Although this percentage is very low, it is approximately 2.5 times higher than that of immigrant children in the same age group in our study. In a different study carried out on school-aged children with a low-socio-economic level in our country, 64.2% of the children had a toothbrush, 33.1% had regular tooth brushing habits [35]. Tooth brushing habits were found to be lower in immigrant children in our study. This may be due to the fact that immigrant children may not have a toothbrush due to economic problems, and due to the low awareness about brushing habits among immigrant children with toothbrushes. Additionally, children whose parents were university graduates and who were in the 6–7 year age group were brushing their teeth at a significantly higher rate, in our study. Moreover, the percentage of tooth brushing was higher in children who had applied to the dentist before, but statistical significance was not observed. In a study on girls, older children and offspring of mothers with higher levels of schooling were more likely to be tooth brushing more frequently [36]. The parental educational levels affect the tooth brushing habit of children in a similar way in our study and the other study in the literature. But different results about age and gender may be caused by socio-demographic differences between study populations such as economic conditions, living place. Besides, in our study, all 15 children without dental caries were brushing their teeth. This result also highlights the importance of tooth brushing. In order to prevent dental caries and improve oral health, educations should be organized to help children to gain oral care behaviors such as tooth brushing, the use of dental floss and interface brushes. Qualitative studies are needed to understand the reasons behind not brushing teeth among immigrant children.

In our study, of the children 61.3% had not previously attended a dental examination, the most common reason for the dentist visit was having dental-oral complaints. Only one child had an annual dental examination without any complaints. In a study conducted in our country, 5.6% of the children aged 11–12 years had never been to the dentist, and 17.6% of them did not remember. Additionally, approximately one fourth of the children went to the dentist for a routine examination without complaints [37]. The percentage of routine dental examination without complaints is also low for dental health, however, higher than results of our study. Dentist examination may be more limited among immigrants due to reasons such as economic problems, transportation difficulties and lack of awareness.

Factors that may be associated with DMFT and dmft scores were evaluated in our study. There were no significant effects on dmft and DMFT scores of gender, child’s own toothbrush, brushing teeth at least once a day, visiting the dentist before and parental educational level. The interesting result that tooth brushing was not a related factor for DMFT and dmft scores can arise from the fact that immigrant children may brush their teeth ineffectively with an inappropriate way and duration. However, the dmft score for the 6–7 year age group was significantly higher than the 8–9 year and 10–12 year age group. In a study conducted in Libya among school-aged children, the prevalence of dental caries was higher in seventh grade students than that of first grade students [13]. This finding supports the idea that dental health is worse at younger ages in school-aged children, similar to our results. The dmft score for children in our study whose parents were university graduates was lower than the others. Although statistical significance was not observed, the *p* value was close to the significance level (*p* = 0.053). In our country, a study conducted among children with high-socio-economic conditions found mean DMFT scores to be 2.7 ± 2.5 in children aged 7–10 years [38]. In the Yeditepe University 2009–2010 study, the DMFT index scores for the 5–9 year age group and 10–14 year age group were reported as 2.8 and 2.2, respectively [39]. Similarly, a study conducted in Greece stated that immigrant children had higher odds for DMFT scores when compared with their Greek counterparts [40]. Studies indicate that the oral and dental health of immigrant children are worse than non-immigrants based on the DMFT index. The DMFT index used in studies help to evaluate and compare the dental health of different groups in a quantitative way. In addition, the planning of qualitative studies on the dental health of immigrant children, in which the clinical experiences and views of dentists are asked, can make a significant contribution to this field.

In our study, 97.3% of the immigrant children (n = 534) had at least one decayed tooth. A study conducted in our country among children aged 7–15 years with disadvantaged socio-economic status reported that 74.8% of the children had dental caries [35]. Whereas in another study conducted in our country among school-aged children with high-socio-economic status, the percentage of children with at least one decayed tooth was reported as 47.1 [38]. Similarly in a study conducted in Spain, immigrant children were found to use the dental health services less frequently and showed a greater risk of dental caries [41]. According to the results of the studies, we can interpret that a low-socio-economic level affects dental health negatively. The higher prevalence of dental caries in children in our study may be due to the fact that immigrants are one of the disadvantaged groups in terms of socio-economic level. Thus, their nutritional conditions, oral care behaviors and applications to the dentist may also be negatively affected.

### Limitations and Strengths

The DMFT and dmft indices provide a way to quantify dental health based on the number of decayed, missing and filled teeth. However, they cannot provide a precise description of prior oral and dental care. This is one of the limitations of this study. In our study, the determination of the dental caries are reported based on clinical examinations without x-rays. It may cause the underestimation of the dental caries. This situation creates another limitation for our study. Additionally, dental screening was carried out only among immigrant children with SIHC applications, thus the study can not have a community-based property. This means that a referral filter bias potentially exists and we cannot generalize our study results, thus creating another limitation. One more limitation is that the factors related to dental caries are not comprehensive in our study. We suggest that there is a need for further exploration of the associated factors for dental caries in future studies.

There are also strengths of our study. In the study, the use of indices for the evaluation of oral and dental health allows for quantitative evaluation. This is the strength of the study. In addition, questions about dental care-related behaviors such as tooth-brushing frequency and flossing, apart from dental caries, provides a broad perspective in the evaluation of the study results and contributes to the strengths of the study. To our knowledge, our study is the unique study on oral-dental health screening in Syrian immigrant children living in our country. This contributes to the strengths of the study.

## 5. Conclusions

### 5.1. Conclusions of the Study

The habit of tooth brushing, which is one of the fundamental requirements of good oral health, remains at a low level among immigrant children in our study. Additionally, dental caries are seen at a very-high frequency. Lower age is found to be related with tooth decay among immigrant children. Although age is not a modifiable risk factor, extra precautions can be taken to prevent dental caries for younger children in clinical practice. The results of the study show that there is a need to improve oral and dental health in immigrant children.

### 5.2. Implications for Future Practice and Research

In order for children to acquire habits related to oral and dental health, interventions can be planned primarily to increase the awareness of individuals who are in contact with the child, such as family members and teachers. Free distribution of oral care-related materials such as toothbrushes can be provided in order to solve the problems arising from economic and transportation problems, especially for disadvantaged children such as immigrants. In addition, dentist examinations can be carried out in schools or households for similar problems that immigrants can face. Health professionals, researchers, educators and policy makers should be encouraged to develop strategies to improve dental health, prevent dental caries and target to solve oral health inequalities in children [42].

## Figures and Tables

**Table 1 children-10-01000-t001:** Socio-demographical characteristics of the Syrian immigrant children.

Features		
Age (years), Median (Min–Max)	7.0 (6.0–12.0)	
Age groups, n (%)	6–7 years	327 (59.6)
	8–9 years	115 (20.9)
	10–12 years	107 (19.5)
Gender, n (%)	Female	276 (50.3)
	Male	273 (49.7)
Parents’ education *, n (%)	Illitarete	75 (13.7)
	Literate	36 (6.6)
	Primary school	207 (37.7)
	Secondary school	156 (28.4)
	High school	44 (8.0)
	Univesity or higher	30 (5.5)

* One parent’s data about education was missing.

**Table 2 children-10-01000-t002:** Characteristics for oral health and oral health habits.

		n (%)
Orthodontic treatment	No	549 (100.0)
	Yes	0 (0.0)
Gingival bleeding	No	519 (94.9)
	Yes	28 (5.1)
Own toothbrush	No	159 (29.1)
	Yes	388 (70.9)
Frequency of tooth brushing	No brushing	155 (28.3)
	Once daily	149 (27.2)
	2-4 times in a day	107 (19.5)
	Irregularly	137 (25.0)
Use of interdental brush or floss	No	545 (99.8)
	Yes	1 (0.2)
Reason for last visit to the dentist	No dental visit before	332 (61.3)
	Pain and complaints	189 (34.9)
	Treatment or follow-up	20 (3.7)
	Annual check-up without complaints	1 (0.2)
The presence of a retainer in the mouth	No	547 (100.0)
	Yes	0 (0.0)
Fluoride varnish application	No	507 (94.6)
	Yes	29 (5.4)

Data were missing for two children for gingival bleeding, having own toothbrush, the presence of a retainer in the mouth, for frequency of tooth brushing in one child, for use of interdental brush or floss in three children, for the reason of the last visit to the dentist in seven children and for fluoride varnish application in thirteen children.

**Table 3 children-10-01000-t003:** Factors related to tooth brushing.

		Tooth Brushing		*p* Value
		No	Yes	
		N (%)	N (%)	
Gender	Female	143 (51.8)	133 (48.2)	0.486
	Male	149 (54.8)	123 (45.2)	
Age group	6–7 years	162 (49.5)	165 (50.5)	0.027
	8–9 years	61 (53.5)	53 (46.5)	
	10–12 years	69 (64.5)	38 (35.5)	
College graduate parent	No	282 (54.7)	234 (45.3)	0.005
	Yes	9 (29.0)	22 (71.0)	
Dentist visit before	No	184 (55.4)	148 (44.6)	0.180
	Yes	104 (49.5)	106 (50.5)	

**Table 4 children-10-01000-t004:** Dental health behaviors and age and gender characteristics of children without caries.

Child	Gender	Age	Own Tooth Brush	Tooth Brushing	Dental Examination before
1.	M	9	Yes	Yes	Yes
2.	M	7	Yes	Yes	No
3.	M	7	Yes	Yes	No
4.	F	7	Yes	Yes	Yes
5.	F	8	Yes	Yes	Yes
6.	F	7	Yes	Yes	No
7.	M	7	Yes	Yes	No
8.	F	6	Yes	Yes	Yes
9.	M	7	Yes	Yes	No
10.	F	9	Yes	Yes	Yes
11.	M	6	Yes	Yes	No
12.	F	7	Yes	Yes	No
13.	F	6	No	No	No
14.	F	8	No	No	Yes
15.	F	7	No	No	Yes

**Table 5 children-10-01000-t005:** DMFT and dmft scores and related factors.

Factors		DMFT	*p* Value	DMFT	*p* Value
		Mean ± SD		Mean ± SD	
Gender	Female	5.30 ± 3.07	0.133	1.05 ± 1.75	0.497
	Male	5.72 ± 3.42		0.95 ± 1.51	
Age group	6–7 years	6.45 ± 3.33	<0.001	1.07 ± 1.73	0.158
	8–9 years	4.98 ± 2.78		1.04 ± 1.66	
	10–12 years	3.22 ± 2.02		0.73 ± 1.25	
Own toothbrush	No	5.60 ± 3.11	0.704	1.15 ± 1.73	0.178
	Yes	5.49 ± 3.31		0.94 ± 1.59	
Tooth brushing more than once a day	No	5.36 ± 3.19	0.240	1.10 ± 1.72	0.150
	Yes	5.69 ± 3.33		0.89 ± 1.52	
Dentist visit before	No	5.60 ± 3.23	0.548	1.04 ± 1.70	0.356
	Yes	5.42 ± 3.29		0.91 ± 1.51	
College graduate parent	No	5.58 ± 3.26	0.053	0.99 ± 1.65	0.725
	Yes	4.42 ± 2.96		1.09 ± 1.35	
Gingival bleeding	No	5.53 ± 3.23	0.398	1.02 ± 1.66	0.470
	Yes	5.00 ± 3.59		0.79 ± 1.20	
Fluoride varnish application	No	5.56 ± 3.23	0.064	0.99 ± 1.61	0.388
	Yes	4.41 ± 3.31		0.72 ± 1.33	

**Table 6 children-10-01000-t006:** Logistic regression analysis for mean dmft score and related factors.

Variables	*p* Value	OR	95% C.I. for OR	
			Lower	Upper
Gender	0.241	1.245	0.863	1.797
Age (6–7 years) *	<0.001	9.863	5.285	18.407
Age (8–9 years) *	<0.001	4.358	2.175	8.732
Fluoride varnish application	0.058	2.345	0.970	5.667

* Reference value for age was 10–12 years age group.

## Data Availability

Not applicable.

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
