# Peer review of "Prevalence and Associated Factors of Dental Caries in Syrian Immigrant Children Aged 6–12 Years"

_children, 2023, doi:10.3390/children10061000_

Round 1
Reviewer 1 Report
Please see the file,thanks.

Most of the content is well written,Minor editing of English language required.
Reviewer 2 Report
Title: Prevalence of Dental Caries and Associated Factors in Syrian Immigrant Children aged 6-12 years
The topic of great interest for the health professionals, however I have few concerns and suggestions
Abstract:
1. The authors have not mentioned about the background of the study
2. The abstract should be return in past tense
3. The authors should mention the findings clearly, what were the DMFT scores?
4. The conclusions of this paper should be based on the findings
Introduction:
1. Introduction is well written however, the authors need to add more details on the prevalence dental caries in this age group. The global scenario should be mentioned.
Methodology:
1. Authors mention about the ethical clearance [IRB] but did not mention regarding the informed consent?
2. How is this a retrospective study?
3. Was the examiner calibrated? Did he have a clinical experience? Is he a part of this research paper?
4. Where was the screening conducted, what about the instrumentation and sterilization?
5. Details of both the indices need to be presented along with the examination charts
6. How did they examine the mixed dentition?
7. What were the inclusion and exclusion criteria’s? Should be mentioned explicitly
Results:
1. The authors mention of gingival bleeding. Was there any index used to measure this?
2. They assessed the brushing frequency for 2-4 times a day. How appropriate is this question?
Discussion:
1. Discussion is very superficial, looks like a continuation of introduction.
2. Results need to be discussed and compared with those reported in literature with proper justification.
3. Should highlight the need for recommendations and policy making based on the results.
4. Limitations of the study design has to be mentioned
5. What are the potential bias that can arise with this type of study design?
6. Can these findings be generalized?
Conclusion:
1. Should be based on the findings
2. It is too general

Reviewer 3 Report
I was pleased to review the manuscript “children-2409543” entitled “Prevalence of Dental Caries and Associated Factors in Syrian Immigrant Children aged 6-12 years” for the Children journal. This is a descriptive and retrospective study in which Syrian immigrant children aged 6-12 years old were screened in Istanbul for oral and dental health. Overall, it is an interesting study, that can bring good social impacts and promote oral health for Syrian immigrants in Turkey. However, major revision is suggested to elucidate some concerns and to improve the content and facilitate the reading.
Please refer to comments below:
Introduction:
The introduction is very long, please try to summarize it a little bit so the reading will be easier.
Materials and Methods:
Do the authors have data about inter- and intra- examiner reliability?
Was a sample size calculation or a post hoc power analyses performed?
The Ethics Committee approved the study only in October/2022, so when were the dental screenings performed (when it started and when ended?)?
Discussion:
The sentence “The 260 majority of children who previously applied to the dentist applied in case of a complaint.” in lines 260-261 is confusing, please rephrase it.
Please take a close look at this whole section because several parts are confusing.
The last paragraph should be rewritten. Try to be more concise regarding the limitations and strengths of the study, it is a little repetitive and confusing.
Conclusions:
The conclusion is a bit too long although it brings good information. The authors can consider moving some of this to the Discussion and try to summarize the Conclusion.
The manuscript will be better presented after editing to reduce the confusion observed in some parts (specially in the Discussion).
Round 2
Reviewer 1 Report
Thanks for the authors. They made lots of revisions, very good. There are some minor suggestions before publication.
1. Line 81, “3.411.029 ” should be “3, 411, 029” please change “1.635.397” in the same way.
2. Table 5, gender P=0.133,Please change the format.
3. Table 6, delete the numbers of lines 250-254.
4. Finally, check english editing and references, data in tables again, ensure they are right.
Thanks again.
The Quality of English Languag is good.
Reviewer 3 Report
The authors attended to the suggestions and provided more details about the methodology, which fulfilled my concerns. The manuscript is better in general, being more clear, better discussed and providing more organized conclusions. Therefore, I recommend its acceptance in the present form.
The writing was improved and the reading is now easier.